# 🍺 BEIR: A Heterogeneous Benchmark for Zero-shot Evaluation of Information Retrieval Models

**Nandan Thakur, Nils Reimers, Andreas Rücklé,** *Abhishek Srivastava, Iryna Gurevych**
Ubiquitous Knowledge Processing Lab (UKP Lab)
Department of Computer Science, Technical University of Darmstadt
http://www.ukp.tu-darmstadt.de

## Abstract

Existing neural information retrieval (IR) models have often been studied in homogeneous and narrow settings, which has considerably limited insights into their out-of-distribution (OOD) generalization capabilities. To address this, and to facilitate researchers to broadly evaluate the effectiveness of their models, we introduce **Be**nchmarking-**IR** (BEIR), a robust and heterogeneous evaluation benchmark for information retrieval. We leverage a careful selection of 18 publicly available datasets from diverse text retrieval tasks and domains and evaluate 10 state-of-the-art retrieval systems including lexical, sparse, dense, late-interaction and re-ranking architectures on the BEIR benchmark. Our results show BM25 is a robust baseline and re-ranking and late-interaction based models on average achieve the best zero-shot performances, however, at high computational costs. In contrast, dense and sparse-retrieval models are computationally more efficient but often underperform other approaches, highlighting the considerable room for improvement in their generalization capabilities. We hope this framework allows us to better evaluate and understand existing retrieval systems, and contributes to accelerating progress towards more robust and generalizable systems in the future. BEIR is publicly available at https://github.com/UKPLab/beir.

## 1 Introduction

Major natural language processing (NLP) problems rely on a practical and efficient retrieval component as a first step to find relevant information. Challenging problems include open-domain question-answering [8], claim-verification [58], duplicate question detection [77], and many more. Traditionally, retrieval has been dominated by lexical approaches like TF-IDF or BM25 [53]. However, these approaches suffer from lexical gap [5] and are able to only retrieve documents containing keywords present within the query. Further, lexical approaches treat queries and documents as bag-of-words by not taking word ordering into consideration.

Recently, deep learning and in particular pre-trained Transformer models like BERT [12] have become popular in information retrieval [75]. These neural retrieval systems can be used in many fundamentally different ways to improve retrieval performance. We provide an brief overview of the systems in Section 2.1. Many prior work train neural retrieval systems on large datasets like Natural Questions (NQ) [32] (133k training examples) or MS MARCO [42] (533k training examples), which both focus on passage retrieval given a question or short keyword-based query. In most prior work, approaches are afterward evaluated on the same dataset, where significant performance gains over lexical approaches like BM25 are demonstrated [48, 29, 43].

However, creating a large training corpus is often time-consuming and expensive and hence many retrieval systems are applied in a **zero-shot setup**, with no available training data to train the system.

---

*Contributions made prior to joining Amazon.

35th Conference on Neural Information Processing Systems (NeurIPS 2021) Track on Datasets and Benchmarks.

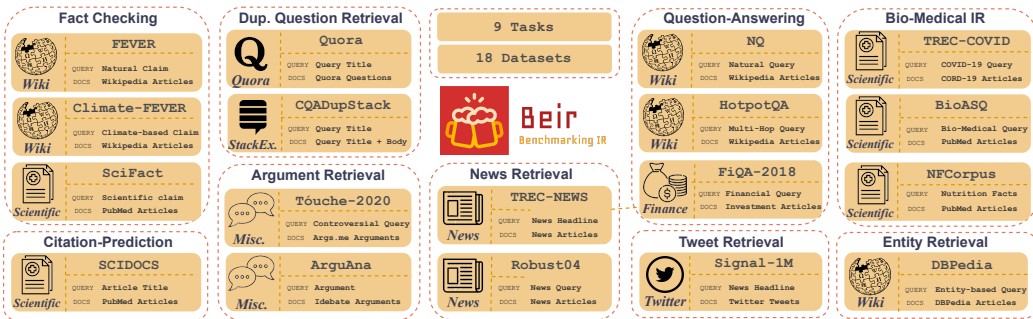

**Figure 1:** An overview of the diverse tasks and datasets in BEIR benchmark.

So far, it is unclear how well existing trained neural models will perform for other text domains or textual retrieval tasks. Even more important, it is unclear how well different approaches, like sparse embeddings vs. dense embeddings, generalize to out-of-distribution data.

In this work, we present a novel robust and heterogeneous benchmark called **BEIR** (**Be**nchmarking **IR**), comprising of 18 retrieval datasets for comparison and evaluation of model generalization. Prior retrieval benchmarks [17, 47] have issues of a comparatively narrow evaluation focusing either only on a single task, like question-answering, or on a certain domain. In BEIR, we focus on **Diversity**, we include nine different retrieval tasks: Fact checking, citation prediction, duplicate question retrieval, argument retrieval, news retrieval, question answering, tweet retrieval, bio-medical IR, and entity retrieval. Further, we include datasets from diverse text domains, datasets that cover broad topics (like Wikipedia) and specialized topics (like COVID-19 publications), different text types (news articles vs. Tweets), datasets of various sizes (3.6k - 15M documents), and datasets with different query lengths (average query length between 3 and 192 words) and document lengths (average document length between 11 and 635 words).

We use BEIR to evaluate **ten diverse retrieval methods** from five broad architectures: lexical, sparse, dense, late interaction, and re-ranking. From our analysis, we find that no single approach consistently outperforms other approaches on all datasets. Further, we notice that the in-domain performance of a model does not correlate well with its generalization capabilities: models fine-tuned with identical training data might generalize differently. In terms of efficiency, we find a trade-off between the performances and the computational cost: computationally expensive models, like re-ranking models and late interaction model perform the best. More efficient approaches e.g. based on dense or sparse embeddings can substantially underperform traditional lexical models like BM25. Overall, BM25 remains a strong baseline for zero-shot text retrieval.

Finally, we notice that there can be a strong lexical bias present in datasets included within the benchmark, likely as lexical models are pre-dominantly used during the annotation or creation of datasets. This can give an unfair disadvantage to non-lexical approaches. We analyze this for the TREC-COVID [63] dataset: We manually annotate the missing relevance judgements for the tested systems and see a significant performance improvement for non-lexical approaches. Hence, future work requires better unbiased datasets that allow a fair comparison for all types of retrieval systems.

With BEIR, we take an important step towards a single and unified benchmark to evaluate the zero-shot capabilities of retrieval systems. It allows to study when and why certain approaches perform well, and hopefully steers innovation to more robust retrieval systems. We release BEIR and an integration of diverse retrieval systems and datasets in a well-documented, easy to use and extensible open-source package. BEIR is model-agnostic, welcomes methods of all kinds, and also allows easy integration of new tasks and datasets. More details are available at https://github.com/UKPLab/beir.

## 2 Related Work and Background

To our knowledge, BEIR is the first broad, zero-shot information retrieval benchmark. Existing works [17, 47] do not evaluate retrieval in a zero-shot setting in depth, they either focus over a single task, small corpora or on a certain domain. This setting hinders for investigation of model generalization across diverse set of domains and task types. MultiReQA [17] consists of eight Question-Answering (QA) datasets and evaluates sentence-level answer retrieval given a question. It only tests a single task and five out of eight datasets are from Wikipedia. Further, MultiReQA evaluates retrieval over rather small corpora: six out of eight tasks have less than 100k candidate sentences, which benefits dense retrieval over lexical as previously shown [52]. KILT [47] consists of five knowledge-intensive

tasks including a total of eleven datasets. The tasks involve retrieval, but it is not the primary task. Further, KILT retrieves documents only from Wikipedia.

## 2.1 Neural Retrieval

Information retrieval is the process of searching and returning relevant documents for a query from a collection. In our paper, we focus on text retrieval and use *document* as a cover term for text of any length in the given collection and *query* for the user input, which can be of any length as well. Traditionally, lexical approaches like TF-IDF and BM25 [53] have dominated textual information retrieval. Recently, there is a strong interest in using neural networks to improve or replace these lexical approaches. In this section, we highlight a few neural-based approaches and we refer the reader to Lin et al. [75] for a recent survey in neural retrieval.

**Retriever-based** Lexical approaches suffer from the lexical gap [5]. To overcome this, earlier techniques proposed to improve lexical retrieval systems with neural networks. Sparse methods such as docT5query [45] identified document expansion terms using a sequence-to-sequence model that generated possible queries for which the given document would be relevant. DeepCT [11] on the other hand used a BERT [12] model to learn relevant term weights in a document and generated a pseudo-document representation. Both methods still rely on BM25 for the remaining parts. Similarly, SPARTA [78] learned token-level contextualized representations with BERT and converted the document into an efficient inverse index. More recently, dense retrieval approaches were proposed. They are capable of capturing semantic matches and try to overcome the (potential) lexical gap. Dense retrievers map queries and documents in a shared, dense vector space [16]. This allowed the document representation to be pre-computed and indexed. A bi-encoder neural architecture based on pre-trained Transformers has shown strong performance for various open-domain question-answering tasks [17, 29, 33, 40]. This dense approach was recently extended by hybrid lexical-dense approaches which aims to combine the strengths of both approaches [15, 55, 39]. Another parallel line of work proposed an unsupervised domain-adaption approach [33, 40] for training dense retrievers by generating synthetic queries on a target domain. Lastly, ColBERT [30] (Contextualized late interaction over BERT) computes multiple contextualized embeddings on a token level for queries and documents and uses an maximum-similarity function for retrieving relevant documents.

**Re-ranking-based** Neural re-ranking approaches use the output of a first-stage retrieval system, often BM25, and re-ranks the documents to create a better comparison of the retrieved documents. Significant improvement in performance was achieved with the cross-attention mechanism of BERT [43]. However, at a disadvantage of a high computational overhead [51].

## 3 The BEIR Benchmark

BEIR aims to provide a one-stop zero-shot evaluation benchmark for all diverse retrieval tasks. To construct a comprehensive evaluation benchmark, the selection methodology is crucial to collect tasks and datasets with desired properties. For BEIR, the methodology is motivated by the following three factors: (*i*) **Diverse tasks**: Information retrieval is a versatile task and the lengths of queries and indexed documents can differ between tasks. Sometimes, queries are short, like a keyword, while in other cases, they can be long like a news article. Similarly, indexed documents can sometimes be long, and for other tasks, short like a tweet. (*ii*) **Diverse domains**: Retrieval systems should be evaluated in various types of domains. From broad ones like News or Wikipedia, to highly specialized ones such as scientific publications in one particular field. Hence, we include domains which provide a representation of real-world problems and are diverse ranging from generic to specialized. (*iii*) **Task difficulties**: Our benchmark is challenging and the *difficulty* of a task included has to be sufficient. If a task is easily solved by any algorithm, it will not be useful to compare various models used for evaluation. We evaluated several tasks based on existing literature and selected popular tasks which we believe are recently developed, challenging and are not yet fully solved with existing approaches. (*iv*) **Diverse annotation strategies**: Creating retrieval datasets are inherently complex and are subject to *annotation biases* (see Section 6 for details), which hinders a fair comparison of approaches. To reduce the impact of such biases, we selected datasets which have been created in many different ways: Some where annotated by crowd-workers, others by experts, and others are based on the feedback from large online communities.

In total, we include 18 English zero-shot evaluation datasets from 9 heterogeneous retrieval tasks. As the majority of the evaluated approaches are trained on the MS MARCO [42] dataset, we also report performances on this dataset, but don't include the outcome in our zero-shot comparison. We would like to refer the reader to Appendix C where we motivate each one of the 9 retrieval tasks and 18

| Split (→) | | | | | Train | Dev | Test | | | Avg. Word Lengths | |
|---|---|---|---|---|---|---|---|---|---|---|---|
| Task (↓) | Domain (↓) | Dataset (↓) | Title | Relevancy | #Pairs | #Query | #Query | #Corpus | Avg. D / Q | Query | Document |
| Passage-Retrieval | Misc. | MS MARCO [42] | ✗ | Binary | 532,761 | —— | 6,980 | 8,841,823 | 1.1 | 5.96 | 55.98 |
| Bio-Medical | Bio-Medical | TREC-COVID [63] | ✓ | 3-level | —— | —— | 50 | 171,332 | 493.5 | 10.60 | 160.77 |
| Information | Bio-Medical | NFCorpus [7] | ✓ | 3-level | 110,575 | 324 | 323 | 3,633 | 38.2 | 3.30 | 232.26 |
| Retrieval (IR) | Bio-Medical | BioASQ [59] | ✓ | Binary | 32,916 | —— | 500 | 14,914,602 | 4.7 | 8.05 | 202.61 |
| Question | Wikipedia | NQ [32] | ✓ | Binary | 132,803 | —— | 3,452 | 2,681,468 | 1.2 | 9.16 | 78.88 |
| Answering | Wikipedia | HotpotQA [74] | ✓ | Binary | 170,000 | 5,447 | 7,405 | 5,233,329 | 2.0 | 17.61 | 46.30 |
| (QA) | Finance | FiQA-2018 [41] | ✗ | Binary | 14,166 | 500 | 648 | 57,638 | 2.6 | 10.77 | 132.32 |
| Tweet-Retrieval | Twitter | Signal-1M (RT) [57] | ✗ | 3-level | —— | —— | 97 | 2,866,316 | 19.6 | 9.30 | 13.93 |
| News | News | TREC-NEWS [56] | ✓ | 5-level | —— | —— | 57 | 594,977 | 19.6 | 11.14 | 634.79 |
| Retrieval | News | Robust04 [62] | ✗ | 3-level | —— | —— | 249 | 528,155 | 69.9 | 15.27 | 466.40 |
| Argument | Misc. | ArguAna [65] | ✓ | Binary | —— | —— | 1,406 | 8,674 | 1.0 | 192.98 | 166.80 |
| Retrieval | Misc. | Touché-2020 [6] | ✓ | 3-level | —— | —— | 49 | 382,545 | 19.0 | 6.55 | 292.37 |
| Duplicate-Question | StackEx. | CQADupStack [23] | ✓ | Binary | —— | —— | 13,145 | 457,199 | 1.4 | 8.59 | 129.09 |
| Retrieval | Quora | Quora | ✗ | Binary | —— | 5,000 | 10,000 | 522,931 | 1.6 | 9.53 | 11.44 |
| Entity-Retrieval | Wikipedia | DBPedia [19] | ✓ | 3-level | —— | 67 | 400 | 4,635,922 | 38.2 | 5.39 | 49.68 |
| Citation-Prediction | Scientific | SCIDOCS [9] | ✓ | Binary | —— | —— | 1,000 | 25,657 | 4.9 | 9.38 | 176.19 |
| Fact Checking | Wikipedia | FEVER [58] | ✓ | Binary | 140,085 | 6,666 | 6,666 | 5,416,568 | 1.2 | 8.13 | 84.76 |
| | Wikipedia | Climate-FEVER [13] | ✓ | Binary | —— | —— | 1,535 | 5,416,593 | 3.0 | 20.13 | 84.76 |
| | Scientific | SciFact [66] | ✓ | Binary | 920 | —— | 300 | 5,183 | 1.1 | 12.37 | 213.63 |

Table 1: **Statistics of datasets** in BEIR benchmark. Few datasets contain documents without titles. Relevancy indicates the query-document relation: binary (relevant, non-relevant) or graded into sub-levels. Avg. D/Q indicates the average relevant documents per query.

datasets in depth. Examples for each dataset are listed in Table 8. We additionally provide dataset licenses in Appendix D, and links to the datasets in Table 5.

Table 1 summarizes the statistics of the datasets provided in BEIR. A majority of datasets contain binary relevancy judgements, i.e. relevant or non-relevant, and a few contain fine-grained relevancy judgements. Some datasets contain few relevant documents for a query (< 2), while other datasets like TREC-COVID [63] can contain up to even 500 relevant documents for a query. Only 8 out of 19 datasets (including MS MARCO) have training data denoting the practical importance for zero-shot retrieval benchmarking. All datasets except ArguAna [65] have short queries (either a single sentence or 2-3 keywords). Figure 1 shows an overview of the tasks and datasets in the BEIR benchmark.

Information Retrieval (IR) is ubiquitous, there are lots of datasets available within each task and further even more tasks with retrieval. However, it is not feasible to include all datasets within the benchmark for evaluation. We tried to cover a balanced mixture of a wide range of tasks and datasets and paid importance not to overweight a specific task like question-answering. Future datasets can easily be integrated in BEIR, and existing models can be evaluated on any new dataset quickly. The BEIR website will host an actively maintained leaderboard[2] with all datasets and models.

## 3.1 Dataset and Diversity Analysis

The datasets present in BEIR are selected from diverse domains ranging from Wikipedia, scientific publications, Twitter, news, to online user communities, and many more. To measure the diversity in domains, we compute the domain overlap between the pairwise datasets using a pairwise weighted Jaccard similarity [24] score on unigram word overlap between all dataset pairs. For more details on the theoretical formulation of the similarity score, please refer to Appendix E. Figure 2 shows a heatmap denoting the pairwise weighted jaccard scores and the clustered force-directed placement diagram. Nodes (or datasets) close in this graph have a high word overlap, while nodes far away in the graph have a low overlap. From Figure 2, we observe a rather low weighted Jaccard word overlap across different domains, indicating that BEIR is a challenging benchmark where approaches must generalize well to diverse out-of-distribution domains.

## 3.2 BEIR Software and Framework

The BEIR software[3] provides an is an easy to use Python framework (`pip install beir`) for model evaluation. It contains extensive wrappers to replicate experiments and evaluate models from well-known repositories including Sentence-Transformers [51], Transformers [70], Anserini [72], DPR [29], Elasticsearch, ColBERT [30], and Universal Sentence Encoder [73]. This makes the software useful for both academia and industry. The software also provides you with all IR-based metrics from Precision, Recall, MAP (Mean Average Precision), MRR (Mean Reciprocal Rate) to nDCG

---

[2]BEIR Leaderboard: https://tinyurl.com/beir-leaderboard

[3]BEIR Code & documentation: https://github.com/UKPLab/beir

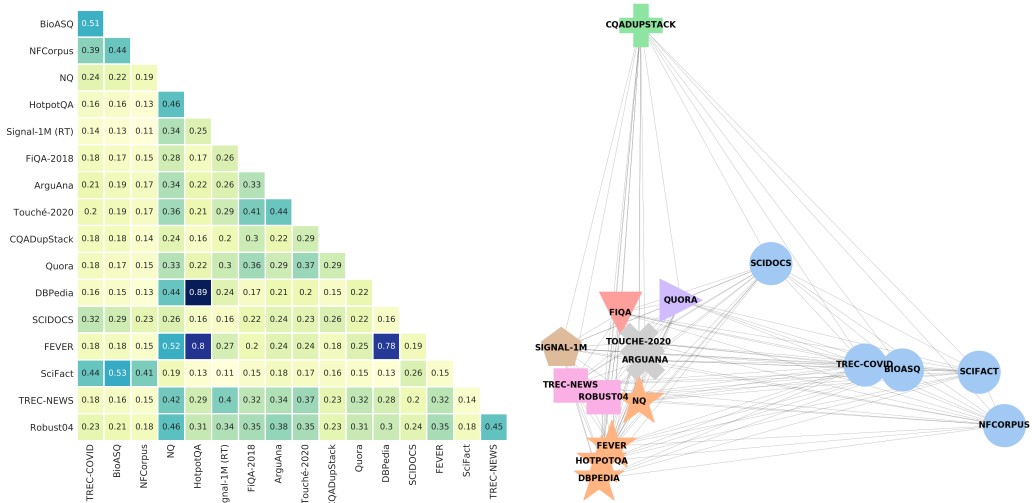

**Figure 2:** Domain overlap across each pairwise dataset in the BEIR benchmark. Heatmap (left) shows the pairwise weighted jaccard similarity scores between BEIR datasets. 2D representation (right) using a force-directed placement algorithm with NetworkX [18]. We color and mark datasets differently for different domains.

(Normalised Cumulative Discount Gain) for any top-k hits. One can use the BEIR benchmark for evaluating existing models on new retrieval datasets and for evaluating new models on the included datasets.

Datasets are often scattered online and are provided in various file-formats, making the evaluation of models on various datasets difficult. BEIR introduces a standard format (corpus, queries and qrels) and converts existing datasets in this easy universal data format, allowing to evaluate faster on an increasing number of datasets.

### 3.3 Evaluation Metric

Depending upon the nature and requirements of real-world applications, retrieval tasks can be either be precision or recall focused. To obtain comparable results across models and datasets in BEIR, we argue that it is important to leverage a single evaluation metric that can be computed comparably across all tasks. Decision support metrics such as Precision and Recall which are both rank unaware are not suitable. Binary rank-aware metrics such as MRR (Mean Reciprocal Rate) and MAP (Mean Average Precision) fail to evaluate tasks with graded relevance judgements. We find that **Normalised Cumulative Discount Gain** (nDCG@k) provides a good balance suitable for both tasks involving binary and graded relevance judgements. We refer the reader to Wang et al. [69] for understanding the theoretical advantages of the metric. For our experiments, we utilize the Python interface of the official TREC evaluation tool [61] and compute nDCG@10 for all datasets.

## 4    Experimental Setup

We use BEIR to compare diverse, recent, state-of-the-art retrieval architectures with a focus on transformer-based neural approaches. We evaluate on publicly available pre-trained checkpoints, which we provide in Table 6. Due to the length limitations of transformer-based networks, we use only the first 512 word pieces within all documents in our experiments across all neural architectures.

We group the models based on their architecture: (*i*) lexical, (*ii*) sparse, (*iii*) dense, (*iv*) late-interaction, and (*v*) re-ranking. Besides the included models, the BEIR benchmark is model agnostic and in future different model configurations can be easily incorporated within the benchmark.

**(*i*) Lexical Retrieval**: (*a*) **BM25** [53] is a commonly-used bag-of-words retrieval function based on token-matching between two high-dimensional sparse vectors with TF-IDF token weights. We use Anserini [34] with the default Lucene parameters (k=0.9 and b=0.4). We index the title (if available) and passage as separate fields for documents. In our leaderboard, we also tested Elasticsearch BM25 and Anserini + RM3 expansion, but found Anserini BM25 to perform the best.

(**ii**) **Sparse Retrieval**: (*a*) **DeepCT** [11] uses a bert-base-uncased model trained on MS MARCO to learn the term weight frequencies (tf). It generates a pseudo-document with keywords multiplied with the learnt term-frequencies. We use the original setup of Dai and Callan [11] in combination with BM25 with default Anserini parameters which we empirically found to perform better over the tuned MS MARCO parameters. (*b*) **SPARTA** [78] computes similarity scores between the non-contextualized query embeddings from BERT with the contextualized document embeddings. These scores can be pre-computed for a given document, which results in a 30k dimensional sparse vector. As the original implementation is not publicly available, we re-implemented the approach. We fine-tune a DistilBERT [54] model on the MS MARCO dataset and use sparse-vectors with 2,000 non-zero entries. (*c*) **DocT5query** [44] is a popular document expansion technique using a T5 (base) [50] model trained on MS MARCO to generate synthetic queries and append them to the original document for lexical search. We replicate the setup of Nogueira and Lin [44] and generate 40 queries for each document and use BM25 with default Anserini parameters.

(**iii**) **Dense Retrieval**: (*a*) **DPR** [29] is a two-tower bi-encoder trained with a single BM25 hard negative and in-batch negatives. We found the open-sourced Multi model to perform better over the single NQ model in our setting. The Multi-DPR model is a bert-base-uncased model trained on four QA datasets (including titles): NQ [32], TriviaQA [28], WebQuestions [4] and CuratedTREC [3]. (*b*) **ANCE** [71] is a bi-encoder constructing hard negatives from an Approximate Nearest Neighbor (ANN) index of the corpus, which in parallel updates to select hard negative training instances during fine-tuning of the model. We use the publicly available RoBERTa [38] model trained on MS MARCO [42] for 600K steps for our experiments. (*c*) **TAS-B** [21] is a bi-encoder trained with Balanced Topic Aware Sampling using dual supervision from a cross-encoder and a ColBERT model. The model was trained with a combination of both a pairwise Margin-MSE [22] loss and an in-batch negative loss function. We use the publicly available DistilBERT [54] model for our experiments. (*d*) **GenQ**: is an unsupervised domain-adaption approach for dense retrieval models by training on synthetically generated data. First, we fine-tune a T5 (base) [50] model on MS MARCO for 2 epochs. Then, for a target dataset we generate 5 queries for each document using a combination of top-k and nucleus-sampling (top-k: 25; top-p: 0.95). Due to resource constraints, we cap the maximum number of target documents in each dataset to 100K. For retrieval, we continue to fine-tune the TAS-B model using in-batch negatives on the synthetic queries and document pair data. Note, GenQ creates an independent model for each task.

(**iv**) **Late-Interaction**: (*a*) **ColBERT** [30] encodes and represents the query and passage into a bag of multiple contextualized token embeddings. The late-interactions are aggregated with sum of the max-pooling query term and a dot-product across all passage terms. We use the ColBERT model as a dense-retriever (end-to-end retrieval as defined [30]): first top-k candidates are retrieved using ANN with faiss [27] (faiss depth = 100) and ColBERT re-ranks by computing the late aggregated interactions. We train a bert-base-uncased model, with maximum sequence length of 300 on the MS MARCO dataset for 300K steps.

(**v**) **Re-ranking model**: (*a*) **BM25 + CE** [68] reranks the top-100 retrieved hits from a first-stage BM25 (Anserini) model. We evaluated 14 different cross-attentional re-ranking models that are publicly available on the HuggingFace model hub and found that a 6-layer, 384-h MiniLM [68] cross-encoder model offers the best performance on MS MARCO. The model was trained on MS MARCO using a knowledge distillation setup with an ensemble of three teacher models: BERT-base, BERT-large, and ALBERT-large models following the setup in Hofstätter et al. [22].

**Training Setup**: The models included for zero-shot evaluation were originally trained differently. DocT5query and DeepCT were trained for document expansion and term re-weighting. Cross encoder (MiniLM) and SPARTA were both trained with ranking data. All dense retrieval models (DPR, ANCE, and TAS-B) and ColBERT [30] were trained with a mixture: ranking data and random in-batch negatives. Another vital difference lies in hard negatives, few models are trained on better optimized hard negatives whereas others using simpler hard negatives, which may suggest an unfair comparison. DPR was trained using the mined BM25 hard negatives, ColBERT with the original MS MARCO [42] provided hard negatives, ANCE with mined approximate hard negatives, whereas TAS-B used a cross-model distillation from a cross-encoder and a ColBERT model together with BM25 hard negatives.

| Model (→) | Lexical | Sparse | | | Dense | | | | Late-Interaction | Re-ranking |
|---|---|---|---|---|---|---|---|---|---|---|
| Dataset (↓) | BM25 | DeepCT | SPARTA | docT5query | DPR | ANCE | TAS-B | GenQ | ColBERT | BM25+CE |
| MS MARCO | 0.228 | 0.296‡ | 0.351‡ | 0.338‡ | 0.177 | 0.388‡ | 0.408‡ | 0.408‡ | **0.425**‡ | 0.413‡ |
| TREC-COVID | 0.656 | 0.406 | 0.538 | 0.713 | 0.332 | 0.654 | 0.481 | 0.619 | 0.677 | **0.757** |
| BioASQ | 0.465 | 0.407 | 0.351 | 0.431 | 0.127 | 0.306 | 0.383 | 0.398 | 0.474 | **0.523** |
| NFCorpus | 0.325 | 0.283 | 0.301 | 0.328 | 0.189 | 0.237 | 0.319 | 0.319 | 0.305 | **0.350** |
| NQ | 0.329 | 0.188 | 0.398 | 0.399 | 0.474‡ | 0.446 | 0.463 | 0.358 | 0.524 | **0.533** |
| HotpotQA | 0.603 | 0.503 | 0.492 | 0.580 | 0.391 | 0.456 | 0.584 | 0.534 | 0.593 | **0.707** |
| FiQA-2018 | 0.236 | 0.191 | 0.198 | 0.291 | 0.112 | 0.295 | 0.300 | 0.308 | 0.317 | **0.347** |
| Signal-1M (RT) | 0.330 | 0.269 | 0.252 | 0.307 | 0.155 | 0.249 | 0.289 | 0.281 | 0.274 | **0.338** |
| TREC-NEWS | 0.398 | 0.220 | 0.258 | 0.420 | 0.161 | 0.382 | 0.377 | 0.396 | 0.393 | **0.431** |
| Robust04 | 0.408 | 0.287 | 0.276 | 0.437 | 0.252 | 0.392 | 0.427 | 0.362 | 0.391 | **0.475** |
| ArguAna | 0.315 | 0.309 | 0.279 | 0.349 | 0.175 | 0.415 | 0.429 | **0.493** | 0.233 | 0.311 |
| Touché-2020 | **0.367** | 0.156 | 0.175 | 0.347 | 0.131 | 0.240 | 0.162 | 0.182 | 0.202 | 0.271 |
| CQADupStack | 0.299 | 0.268 | 0.257 | 0.325 | 0.153 | 0.296 | 0.314 | 0.347 | 0.350 | **0.370** |
| Quora | 0.789 | 0.691 | 0.630 | 0.802 | 0.248 | 0.852 | 0.835 | 0.830 | **0.854** | 0.825 |
| DBPedia | 0.313 | 0.177 | 0.314 | 0.331 | 0.263 | 0.281 | 0.384 | 0.328 | 0.392 | **0.409** |
| SCIDOCS | 0.158 | 0.124 | 0.126 | 0.162 | 0.077 | 0.122 | 0.149 | 0.143 | 0.145 | **0.166** |
| FEVER | 0.753 | 0.353 | 0.596 | 0.714 | 0.562 | 0.669 | 0.700 | 0.669 | 0.771 | **0.819** |
| Climate-FEVER | 0.213 | 0.066 | 0.082 | 0.201 | 0.148 | 0.198 | 0.228 | 0.175 | 0.184 | **0.253** |
| SciFact | 0.665 | 0.630 | 0.582 | 0.675 | 0.318 | 0.507 | 0.643 | 0.644 | 0.671 | **0.688** |
| Avg. Performance vs. BM25 | | - 27.9% | - 20.3% | + 1.6% | - 47.7% | - 7.4% | - 2.8% | - 3.6% | + 2.5% | + 11% |

**Table 2:** In-domain and zero-shot performances on BEIR benchmark. All scores denote **nDCG@10**. The best score on a given dataset is marked in **bold**, and the second best is underlined. Corresponding Recall@100 performances can be found in Table 9. ‡ indicates the in-domain performances.

## 5   Results and Analysis

In this section, we evaluate and analyze how retrieval models perform on the BEIR benchmark. Table 2 reports the results of all evaluated systems on the selected benchmark datasets. As a baseline, we compare our retrieval systems against BM25. Figure 3 shows, on how many datasets a respective model is able to perform better or worse than BM25.

**1. In-domain performance is not a good indicator for out-of-domain generalization.** We observe BM25 heavily underperforms neural approaches by 7-18 points on in-domain MS MARCO. However, BEIR reveals it to be a strong baseline for generalization and generally outperforming many other, more complex approaches. This stresses the point, that retrieval methods must be evaluated on a broad range of datasets.

**2. Term-weighting fails, document expansion captures out-of-domain keyword vocabulary.** DeepCT and SPARTA both use a transformer network to learn term weighting. While both methods perform well in-domain on MS MARCO, they completely fail to generalize well by under performing BM25 on nearly all datasets. In contrast, document expansion based docT5query is able to add new relevant keywords to a document and performs strong on the BEIR datasets. It outperforms BM25 on 11/18 datasets while providing a competitive performance on the remaining datasets.

**3. Dense retrieval models with issues for out-of-distribution data.** Dense retrieval models (esp. ANCE and TAS-B), that map queries and documents independently to vector spaces, perform strongly on certain datasets, while on many other datasets perform significantly worse than BM25. For example, dense retrievers are observed to underperform on datasets with a large domain shift compared from what they have been trained on, like in BioASQ, or task-shifts like in Touché-2020. DPR, the only non-MSMARCO trained dataset overall performs the worst in generalization on the benchmark.

**4. Re-ranking and Late-Interaction models generalize well to out-of-distribution data.** The cross-attentional re-ranking model (BM25+CE) performs the best and is able to outperform BM25 on almost all (16/18) datasets. It only fails on ArguAna and Touché-2020, two retrieval tasks that are extremely different to the MS MARCO training dataset. The late-interaction model ColBERT computes token embeddings independently for the query and document, and scores (query, document)-pairs by a cross-attentional like MaxSim operation. It performs a bit weaker than the cross-attentional re-ranking model, but is still able to outperform BM25 on 9/18 datasets. It appears that cross-attention and cross-attentional like operations are important for a good out-of-distribution generalization.

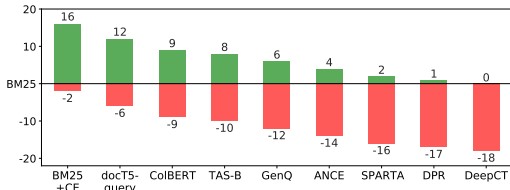

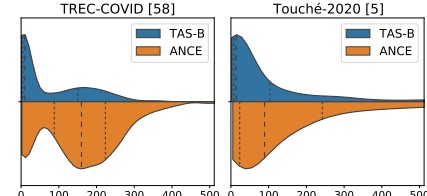

**Figure 3:** Comparison of zero-shot neural retrieval performances with BM25. Re-ranking based models, i.e., BM25+CE and sparse model: docT5query outperform BM25 on more than half the BEIR evaluation datasets.

**Figure 4:** Distribution plots [20] for top-10 retrieved document lengths (in words) using TAS-B (blue, top) or ANCE (orange, bottom). TAS-B has a preference towards shorter documents in BEIR.

**5. Strong training losses for dense retrieval leads to better out-of-distribution performances.** TAS-B provides the best zero-shot generalization performance among its dense counterparts. It outperforms ANCE on 14/18 and DPR on 17/18 datasets respectively. We speculate that the reason lies in a strong training setup in combination of both in-domain batch negatives and Margin-MSE losses for the TAS-B model. This training loss function (with strong ensemble teachers in a Knowledge Distillation setup) shows strong generalization performances.

**6. TAS-B model prefers to retrieve documents with shorter lengths.** TAS-B underperforms ANCE on two datasets: TREC-COVID by 17.3 points and Touché-2020 by 7.8 points. We observed that these models retrieve documents with vastly different lengths as shown in Figure 4. On TREC-COVID, TAS-B retrieves documents with a median length of mere 10 words versus ANCE with 160 words. Similarly on Touché-2020, 14 words vs. 89 words with TAS-B and ANCE respectively. As discussed in Appendix G, this preference for shorter or longer documents is due to the used loss function.

**7. Does domain adaptation help improve generalization of dense-retrievers?** We evaluated GenQ, which further fine-tunes the TAS-B model on synthetic query data. It outperforms the TAS-B model on specialized domains like scientific publications, finance or StackExchange. On broader and more generic domains, like Wikipedia, it performs weaker than the original TAS-B model.

### 5.1 Efficiency: Retrieval Latency and Index Sizes

Models need to potentially compare a single query against millions of documents at inference, hence, a high computational speed for retrieving results in real-time is desired. Besides speed, index sizes are vital and are often stored entirely in memory. We randomly sample 1 million documents from DBPedia [19] and evaluate latency. For dense models, we use exact search, while for ColBERT we follow the original setup [30] and use approximate nearest neighbor search. Performances on CPU were measured with an 8 core Intel Xeon Platinum 8168 CPU @ 2.70GHz and on GPU using a single Nvidia Tesla V100, CUDA 11.0.

**Tradeoff between performance and retrieval latency** The best out-of-distribution generalization performances by re-ranking top-100 BM25 documents and with late-interaction models come at the cost of high latency (> 350 ms), being slowest at inference. In contrast, dense retrievers are 20-30x faster (< 20ms) compared to the re-ranking models and follow a low-latency pattern. On CPU, the sparse models dominate in terms of speed (20-25ms).

| DBPedia [19] (1 Million) | | | Retrieval Latency | | Index |
|---|---|---|---|---|---|
| Rank | Model | Dim. | GPU | CPU | Size |
| (1) | BM25+CE | – | 450ms | 6100ms | 0.4GB |
| (2) | ColBERT | 128 | 350ms | – | 20GB |
| (3) | docT5query | – | – | 30ms | 0.4GB |
| (4) | BM25 | – | – | 20ms | 0.4GB |
| (5) | TAS-B | 768 | 14ms | 125ms | 3GB |
| (6) | GenQ | 768 | 14ms | 125ms | 3GB |
| (7) | ANCE | 768 | 20ms | 275ms | 3GB |
| (8) | SPARTA | 2000 | – | 20ms | 12GB |
| (9) | DeepCT | – | – | 25ms | 0.4GB |
| (10) | DPR | 768 | 19ms | 230ms | 3GB |

**Table 3:** Estimated average retrieval latency and index sizes for a single query in DBPedia [19]. Ranked from best to worst on zero-shot BEIR. Lower the latency or memory is desired.

**Tradeoff between performance and index sizes** Lexical, re-ranking and dense methods have the smallest index sizes (< 3GB) to store 1M documents from DBPedia. SPARTA requires the second largest index to store a 30k dim sparse vector while ColBERT requires the largest index as it stores multiple 128 dim dense vectors for a single document. Index sizes are especially relevant when document sizes scale higher: ColBERT requires ~900GB to store the BioASQ (~15M documents) index, whereas BM25 only requires 18GB.

| Model (→) | BM25 | DeepCT | SPARTA | docT5query | DPR | ANCE | TAS-B | ColBERT | BM25+CE |
|---|---|---|---|---|---|---|---|---|---|
| Hole@10 (in %) | 6.4% | 19.4% | 12.4% | 2.8% | 30.6% | 14.4% | 31.8% | 12.4% | 1.6% |
| nDCG@10 performances before and after manual annotation on TREC-COVID [63] | | | | | | | | | |
| Original (w/ holes) | 0.656 | 0.406 | 0.538 | 0.713 | 0.332 | 0.654 | 0.481 | 0.677 | **0.757** |
| Annotated (w/o holes) | 0.668 | 0.472 | 0.624 | 0.714 | 0.445 | 0.735 | 0.555 | 0.735 | **0.760** |

**Table 4:** Hole@10 analysis on TREC-COVID. Annotated scores show improvement in performances after removing holes@10 (documents in top-10 hits unseen by annotators) across each model.

## 6 Impact of Annotation Selection Bias

Creating a perfectly unbiased evaluation dataset for retrieval is inherently complex and is subject to multiple biases induced by the: (*i*) annotation guidelines, (*ii*) annotation setup, and by the (*iii*) human annotators. Further, it is impossible to manually annotate the relevance for all (query, document)-pairs. Instead, existing retrieval methods are used to get a pool of candidate documents which are then marked for their relevance. All other unseen documents are assumed to be irrelevant. This is a source for *selection bias* [36]: A new retrieval system might retrieve vastly different results than the system used for the annotation. These hits are automatically assumed to be irrelevant.

Many BEIR datasets are found to be subject to a lexical bias, i.e. a lexical based retrieval system like TF-IDF or BM25 has been used to retrieve the candidates for annotation. For example, in BioASQ, candidates have been retrieved for annotation via term-matching with boosting tags [59]. Creation of Signal-1M (RT) involved retrieving tweets for a query with 7 out of these 8 techniques relying upon lexical term-matching signals [57]. Such a lexical bias disfavours approaches that don't rely on lexical matching, like dense retrieval methods, as retrieved hits without lexical overlap are automatically assumed to be irrelevant, even though the hits might be relevant for a query.

In order to study the impact of this particular type of bias, we conducted a study on the recent TREC-COVID dataset. TREC-COVID used a pooling method [35, 37] to reduce the impact of the aforementioned bias: The annotation set was constructed by using the search results from the various systems participating in the challenge. Table 4 shows the Hole@10 rate [71] for the tested systems, i.e., how many top-10 hits is each system retrieving that have not been seen by annotators.

The results reveal large differences between approaches: Lexical approaches like BM25 and docT5query have a rather low Hole@10 value of 6.4% and 2.8%, indicating that the annotation pool contained the top-hits from lexical retrieval systems. In contrast, dense retrieval systems like ANCE and TAS-B have a much higher Hole@10 of 14.4% and 31.8%, indicating that a large fraction of hits found by these systems have not been judged by annotators. Next, we manually added for all systems, the missing annotation (or holes) following the original annotation guidelines. During annotation, we were unaware of the system who retrieved the missing annotation to avoid a preference bias. In total, we annotated 980 query-document pairs in TREC-COVID. We then re-computed nDCG@10 for all systems with this additional annotations.

As shown in Table 4, we observe that lexical approaches improves only slightly, e.g. for docT5query just from 0.713 to 0.714 after adding the missing relevance judgements. In contrast, for the dense retrieval system ANCE, the performance improves from 0.654 (slightly below BM25) to 0.735, which is 6.7 points above the BM25 performance. Similar improvements are noticed in ColBERT (5.8 points). Even though many systems contributed to the TREC-COVID annotation pool, the annotation pool is still biased towards lexical approaches.

## 7 Conclusions and Future Work

In this work, we presented BEIR: a heterogeneous benchmark for information retrieval. We provided a broader selection of target tasks ranging from narrow expert domains to open domain datasets. We included nine different retrieval tasks spanning 18 diverse datasets.

By open-sourcing BEIR, with a standardized data format and easy-to-adapt code examples for many different retrieval strategies, we take an important steps towards a unified benchmark to evaluate the zero-shot capabilities of retrieval systems. It hopefully steers innovation towards more robust retrieval systems and to new insights which retrieval architectures perform well across tasks and domains.

We studied the effectiveness of ten different retrieval models and demonstrate that in-domain performance cannot predict how well an approach will generalize in a zero-shot setup. Many approaches

that outperform BM25 in an in-domain evaluation on MS MARCO, perform poorly on the BEIR datasets. Cross-attentional re-ranking, late-interaction ColBERT, and the document expansion technique docT5query performed overall well across the evaluated tasks.

Our study of the annotation selection bias highlights the challenge of evaluating new models on existing datasets: Even though TREC-COVID is based on the predictions from many systems, contributed by a diverse set of teams, we found largely different *Hole@10* rates for the tested systems, negatively affecting non-lexical approaches. Better datasets that use diverse pooling strategies are needed for a fair evaluation of retrieval approaches. By integrating a large number of diverse retrieval systems into BEIR, creating such diverse pools becomes significantly simplified.

## 8 Limitations of the BEIR Benchmark

Even though we cover a wide range of tasks and domains in BEIR, no benchmark is perfect and has its limitations. Making those explicit is a critical point in understanding the results on the benchmark and, for future work, to propose even better benchmarks.

**1. Multilingual Tasks:** Although we aim for a diverse retrieval evaluation benchmark, due to the limited availability of multilingual retrieval datasets, all datasets covered in the BEIR benchmark are currently English. It is worthwhile to add more multilingual datasets [2, 76] (in consideration of the selection criteria) as a next step for the benchmark. Future work could include multi- and cross-lingual tasks and models.

**2. Long Document Retrieval:** Most of our tasks have average document lengths up-to a few hundred words roughly equivalent to a few paragraphs. Including tasks that require the retrieval of longer documents would be highly relevant. However, as transformer-based approaches often have a length limit of 512 word pieces, a fundamental different setup would be required to compare approaches.

**3. Multi-factor Search:** Until now, we focused on pure textual search in BEIR. In many real-world applications, further signals are used to estimate the relevancy of documents, such as PageRank [46], recency [14], authority score [31] or user-interactions such as click-through rates [49]. The integration of such signals in the tested approaches is often not straight-forward and is an interesting direction for research.

**4. Multi-field Retrieval:** Retrieval can often be performed over multiple fields. For example, for scientific publication we have the title, the abstract, the document body, the authors list, and the journal name. So far we focused only on datasets that have one or two fields.

**5. Task-specific Models:** In our benchmark, we focus on evaluating models that are able to generalize well for a broad range of retrieval tasks. Naturally in real-world, for some few tasks or domains, specialized models are available which can easily outperform generic models as they focus and perform well on a single task, lets say on question-answering. Such task-specific models do not necessarily need to generalize across all diverse tasks.

## 9 Acknowledgements

This work has been supported by the German Research Foundation (DFG) as part of the QASciInf project (grant GU 798/18-3), and the UKP SQuARE project (grant GU 798/29-1). The work has been funded by the German Federal Ministry of Education and Research and the Hessian Ministry of Higher Education, Research, Science and the Arts within their joint support of the National Research Center for Applied Cybersecurity ATHENE. We would like to thank Kexin Wang, Tim Baumgärtner, Leonardo Riberio, Luke Bates, Jan Buchmann for their helpful feedback and participation in the weekly research meetings. Additionally, we would like to thank Chenyan Xiong, Christopher Potts and Sean Macavenay for their constructive feedback and suggestions on Twitter.

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
