# OpenReview forum: "BEIR: A Heterogeneous Benchmark for Zero-shot Evaluation of Information Retrieval Models"
_NeurIPS.cc/2021/Track/Datasets_and_Benchmarks/Round2 — NeurIPS 2021 Datasets and Benchmarks Track (Round 2)_

### Official Review · Reviewer_EhxE · 2021-09-13
**Outstanding new benchmark, with model evaluations that might be more limited than described**

**Rating:** 9
**Confidence:** 3

**Strengths:**

I am confident that BEIR will prove to be a useful benchmark. My collaborators and I have already begun to use it as an evaluation tool.

Although BEIR pools existing datasets, the project is offering much more than that -- the authors have made it very easy to use these datasets to conduct experiments, and they have done a lot of great work in terms of facilitating access to different IR models.

I also really appreciate the "Selection bias" experiments in section 8, which reveal biases towards lexical items across many datasets. This is very useful to know about and it is the sort of thing that the community should be working hard to expose and address.

**Weaknesses:**

I think the model evaluations in the paper  (sections 4 and 5) are a strength on balance, as I said above. However, there are a number of aspects to the results that limit the usefulness of the findings.

* My biggest concern traces to the fact that what we have here are evaluations of artifacts not architectures. In particular, the neural models were trained on substantially different datasets. I think it is clear that MS MARCO is a very different IR resource than, for example, Natural Questions. MS MARCO looks like a reasonable chunk of the useful parts of the Web, whereas NQ is biased by design towards frequently asked questions and so covers much less of the space. Thus, when we compare, for example, DPR with TAS-B, we are not really comparing these as architectures, but rather particular instances of the architecture, and training them on the same data might reduce or remove apparent differences. That's all fair and even useful, but it isn't enough evidence for us to conclude that DPR, as a modeling idea, is different from TAS-B at that level.

* I feel unsure of the category "BM25 + CE". The current paper reports that this is based on MiniLM, but that MiniLM was chosen from a wide range of different models in this category, using MS MARCO as a dev set (if I understand correctly). However, I believe an earlier version of this paper used the model from Nogueira & Cho 2020, and the results table for "BM25 + CE" looked substantially different in that draft (https://arxiv.org/pdf/2104.08663v1.pdf) compared to the current one. So: for "BM25 + CE", it seems like we're seeing the outcome more more dev-set comparisons than for any other model evaluated. Should this affect the conclusions? I am keen to hear more from the authors about this. What is the variance among these different  "BM25 + CE" instantiations?

* I think it's great to assess models with regard to latency and index size, but what about training efficiency? This is also important, and the neural models differ very widely in this regard.

**Additional Feedback:**

Nothing to add here. The primary things I want to discuss are given in the 'Weaknesses' section above.

**Clarity:**

The paper is very clear. I'd urge the authors to proofread it more times, and take note that, by standard English punctuation conventions, a comma cannot join two free-standing sentences unless there is a coordinator like 'and' (and adverbs like 'however' don't count for this).

**Correctness:**

The correctness issue on my mind relate to the scope of the modeling claims, as detailed above.

**Documentation:**

The project comes with pip installable code, and the Github repo contains a lot of good overview scripts and other documentation. It also links out to all the constituent datasets. Excellent stuff all around!


**Ethics:**

No ethical issues in my view. BEIR is pooling lots of others' work, and the authors are careful to cite everyone who should be cited. We have to hope that the community remains this conscientious as time goes on!

**Relation To Prior Work:**

I think the paper does a great job connecting with existing relevant work.


**Summary And Contributions:**

This paper reports on BEIR, a new benchmark based in 18 existing public datasets for information retrieval. BEIR  harmonizes these datasets into a single unified benchmark, mostly oriented towards zero-shot evaluation. The paper provides a basic overview of the BEIR dataset and includes analyses of how the constituent datasets compare with each other. The papers also evaluates a range of different IR models, including term-based models and a variety of models based on distributed/contextual representations. The paper also includes a new analyses providing evidence that the BEIR datasets are biased toward lexical models, and that this bias persists to some extent even where the dataset annotation methods sought to address such biases.

---

> ### Comment · Reviewer_EhxE · 2021-09-29
> **Pressing concerns about the modeling results reported**
>
> I am worried that my high score and overall supportive tone might be obscuring the fact that I have real concerns about the modeling results that need to be addressed. I have lowered my score for now to help convey this.

---

> > ### Author Response · Authors · 2021-09-29
> > **We hope we could address your concerns**
> >
> > We hope we were able to address your concerns with our comment above. Please let us know if you have concerns left so that we can address these here. We also looking forward to hear how we could address these in an updated paper.

---

> > > ### Comment · Reviewer_EhxE · 2021-09-29
> > > **My pressing concern restated**
> > >
> > > I'm afraid my central concern is not really addressed by the above response. My central concern is that some columns in the results table were tuned (in any extended sense) extensively whereas others were not. The extreme case: the column that the authors are sort of arguing for – BM25+CE – is the result of 14 different models all trained specifically for this project, with the best one chosen (based, it seems, at least in part on test set performance, as evidenced by v1 vs the current version). By contrast, it seems like almost all of the other models were tested more or less off-the-shelf.
> > >
> > > What caused the ColBERT and GenQ numbers to drop for this version vs. v1 even while other models were completely unchanged? The BM25 numbers seems also to have shifted around. The BM25+CE column seems to have seen near uniform improvements.
> > >
> > > Surely there is *some version of* the DPR-like single vector approach that will be way better than what is reported here. Why does DPR get only one chance?
> > >
> > > I really want this paper to be published, as I said, but I'm pretty puzzled by the modeling reports.

---

> > > > ### Author Response · Authors · 2021-09-29
> > > > **Explanation of changes - Mainly due to different BM25 algorithm**
> > > >
> > > > Thanks for raising these points.
> > > >
> > > > **Different BM25 algorithm**
> > > >
> > > > A change to the first version on Arxiv is the change of BM25. In the first version of BEIR, we used Elasticsearch for BM25. However, Elasticsearch has the issue that results are not reproducible and can change between versions of Elasticsearch. Based on the recommendation from Jimmy Lin,  we switched to Pyserini (https://github.com/castorini/pyserini/) to ensure reproducibility. BM25 from Pyserini yields better results for most datasets than Elasticsearch, e.g. nDCG@10 for MSMARCO 0.218 -> 0.228, TREC-COVID 0.616 -> 0.656, NFCorpus 0.297 -> 0.325.
> > > >
> > > > As we compare performances against BM25, some results also changed, e.g. on how many datasets is ColBERT better than BM25.
> > > >
> > > > The choice of using Pyserini over Elasticsearch was made solely on the issue of reproducibility of Elasticsearch and was made before computing scores for any of the datasets.
> > > >
> > > > > "is the result of 14 different models all trained specifically for this project, with the best one chosen (based, it seems, at least in part on test set performance, as evidenced by v1 vs the current version)"
> > > >
> > > > The new cross-encoder models based on MarginMSE were **not trained for this project**. They were trained after the MarginMSE paper [3] had been published showing significantly better results on MS MARCO compared to models trained with the training procedure of Nogueira & Cho. On MS MARCO, the MarginMSE model achieves a performance of 39.01 (MRR@10), while the previous model just achieved 36.41 (MRR@10). Following the principle of testing the (available) models with the best MS MARCO dev performance we decided to include this model instead of the previous cross-encoder model. Whenever better versions of certain models are available, we are happy to include them, especially when they achieve such a large in-domain improvement.
> > > >
> > > > Overall, the conclusions had not changed much from version 1: BM25+CE also showed in v1 of the paper the best overall performance, but is extremely compute intensive.
> > > >
> > > > **Elasticsearch + CE vs. Pyserini + CE**
> > > >
> > > > As mentioned,  in the paper we use Pyserini for BM25 and report Pyserini+CE scores. While Pyserini achieves better results on the datasets, Elasticsearch+CE performed actually better than Pyserini+CE. But due to or policy to make model choices before computing performances in BEIR, we still report Psyerini+CE in the paper.
> > > >
> > > >
> > > > **What caused the ColBERT and GenQ numbers to drop for this version vs. v1**
> > > >
> > > > For ColBERT only the performance on Tóuche-2020 changed (0.275 -> 0.202). The authors of Tóuche-2020 provided us new (improved) annotation data. We re-ran all models on these improved annotation data and updated the scores. Note, due to the change in the BM25 algorithm, the relative performance of ColBERT vs BM25 changed in favor of BM25. ColBERT was better on 11 datasets versus BM25-Elasticsearch (version 1), but in this version it is only better on 9 datasets versus BM25-Pyserini.
> > > >
> > > > For GenQ: In this version, GenQ is based on the TAS-B model [2], i.e. we generated queries and continued training the TAS-B model with these generated queries. In the first version of this paper, GenQ was based on an inferior model from SBERT. The better starting point improved GenQ performance.
> > > >
> > > > The decision to switch to TAS-B for a starting point of GenQ was made before training the model, to be consistent with our model selection criteria (select the model with the best MS MARCO performance).
> > > >
> > > >
> > > >
> > > > **Surely there is some version of the DPR-like single vector approach that will be way better than what is reported here. Why does DPR get only one chance?**
> > > > ANCE and TAS-B are actually really similar to DPR: All three models use a transformer network, they all use the CLS token as single vector representation for the query / document, they perform maximum inner product search to find relevant hits for a given query, and ANCE and DPR use the same loss function.
> > > >
> > > > The differences between these is just the training:
> > > > - Different corpora: DPR used multiple QA-datasets, while ANCE & TAS-B used MS MARCO
> > > > - Different base models: roberta-base (ANCE), TAS-B (distilbert), DPR (bert-base)
> > > > - Different training procedures: DPR used BM25 hard negatives, ANCE mined approximate hard negatives, TAS-B uses cross-model distillation from a CrossEncoder and a ColBERT model together with BM25 hard negatives and MarginMSE loss
> > > > - DPR was trained on passages with up-to 100 tokens, while TAS-B & ANCE used passage lengths as provided (on average 56 words)
> > > >
> > > > TAS-B / ANCE perform quite well on certain datasets for a single vector approach, while DPR has issues to generalize. Sadly we cannot say what specific difference made DPR perform significantly worse than TAS-B / ANCE.
> > > >
> > > > But TAS-B and ANCE show that other single vector approaches can perform well.
> > > >
> > > >
> > > > Let us know if you have further questions or concerns.
> > > >
> > > >
> > > > [1] https://arxiv.org/abs/1901.04085
> > > >
> > > > [2] https://arxiv.org/abs/2104.06967
> > > >
> > > > [3] https://arxiv.org/abs/2010.02666

---

> > > > > ### Comment · Reviewer_EhxE · 2021-09-29
> > > > > **Thank you for all these details!**
> > > > >
> > > > > Thank you for all these details! I really appreciate how open the project is and how much effort has clearly gone into reproducibility. I've raised my score back up. I do think we still disagree about the nature of these comparisons, though. To my mind, there are implicitly 14 columns for BM25+CE, not 1, but the "winning" nature of "BM25+CE" is a big theme of the paper. I suppose individual proponents of the other systems that received less attention can use BEIR to argue their case.

---

> > > > > > ### Author Response · Authors · 2021-09-30
> > > > > > **Thanks**
> > > > > >
> > > > > > Thanks for your feedback, we really appreciate that you raised this concern.
> > > > > >
> > > > > > We see your point, that this is a concern and looks like an unfair comparison. We will make it more clear in the paper.
> > > > > >
> > > > > > In fact, in the leaderboard, there are just 8 Cross-Encoders, the other rows are different BM25 configurations (Elasticsearch, Anserini, Anserini+RM3) which we provide for comparison reasons.
> > > > > > Out of the 8 cross-encoders, 4 are really tiny models with only 2 or 4 layers: You don't expect state-of-the-art performance from e.g. a 17 MB model with 2 layers. But the high inference speed and small model size makes it attractive for resource constrained scenarios. Hence, we included it in the leaderboard.
> > > > > >
> > > > > > We will add these points to the final paper that there have been 4 (decently sized) cross-encoders (1 Electra model trained with pointwise loss and 3 MarginMSE trained models) and 4 tiny models (with 2 & 4 layers) to make it transparent to the readers.

---

> ### Author Response · Authors · 2021-09-29
> **Comparing models vs learning approaches**
>
> Thanks for your insightful feedback.
>
> 1) **Comparing models vs approaches**: We agree, the comparison of e.g. TAS-B and DPR is difficult due to the usage of different training datasets. Here, we can just compare the final model, not how well the training procedure works. All other approaches have solely been trained on the MS MARCO passages dataset, which makes a comparison of the training procedures easier. We will make this more visible in the paper, that the presented numbers just compare the available models, but not necessarily the training approach.  Comparing the different training strategies in a fair and equal setup would be highly interesting, but it was sadly not feasible as of now. One reason is that some authors just provide the models without the accompanying training code. But it would be great to see future work that compares training procedures in an equal setup (equal training setup and equal compute budget).
> 2) **Model changes to v1 of the paper:** The first BEIR paper used an Electra model for BM25+CE following the training procedure by Nogueira & Cho [1]. For this version of the paper, we decided to extend the benchmarked models and added docT5query and DeepCT. Further, we added the most recent state-of-the-art models: TAS-B [2] as dense embedding model. For the Cross-Encoder, we replaced the model by a Cross-Encoder trained using MarginMSE [3] as it has been shown superior on MS MARCO compared to the training procedure by Nogueira & Cho. The selection of these two new models (TAS-B and the MarginMSE-base cross-encoder) have been made based on the MS MARCO performance. In principle, we included in the paper the systems with the best MS MARCO dev performance in the respective categories. In the leaderboard (https://tinyurl.com/beir-leaderboard) we also report scores for other models which we could not include in the paper due to space limitations. The update of the Cross-Encoder did not change any conclusions: The previous Cross-Encoder, based on the training procedure from Nogueira & Cho, also outperformed BM25 on most of the datasets and was the best tested system. The improved training with MarginMSE improved the scores a bit further. The same is true for TAS-B, which was also trained with MarginMSE: TAS-B not only performed better in-domain on MS MARCO, but also performed better than previous dense retrieval approaches.
> 3) **Efficiency:** We totally agree that training efficiency largely varies. Even more, indexing speed largely varies between the approaches (e.g. docT5query is about 300 times slower than TAS-B). We hope we can address these points in a future paper to find approaches that provide the best trade-off between performance, latency, index size, indexing speed, and training efficiency.
>
>
> [1] https://arxiv.org/abs/1901.04085
>
> [2] https://arxiv.org/abs/2104.06967
>
> [3] https://arxiv.org/abs/2010.02666

---

### Official Review · Reviewer_cHxA · 2021-09-16
**Very useful benchmark with a few minor issues**

**Rating:** 8
**Confidence:** 4

**Strengths:**

The benchmark they share is huge, covering several datasets and tasks. Therefore, it is useful for zero-shot evaluation.
They provide evaluation results for several IR methods and discuss their performance.
The API they developed seems to be useful.
They share code for their evaluation, which will make it easier to use the benchmark.
The paper is well-written


**Weaknesses:**

I do not see any major weakness in the paper. The authors also provide a list of limitations. One might consider that, authors use existing datasets and pre-trained models for evaluation. So their contribution is limited. I partially agree on that, but their main contribution is easing use of existing datasets. I believe it will be useful for future research.

As another minor weakness, they report only nDCG@10 for their experimental results. It is not clear to me why they picked the cut-off value as 10. To my knowledge, nDCG@10 is not a common metric. For instance, nDCG@20 might be more common.

As another minor issue, using a Google Spreadsheet for leaderboard will be hard to maintain. It would be great if people can just submit their outputs (i.e., output of their systems) and the system will automatically update the leaderboard.


**Additional Feedback:**

The authors convinced me that nDCG@10 is also a reasonable choice to report. Therefore, you can ignore it as one of the weaknesses I raised (I keep it there to understand their answer for that issue)

**Clarity:**

The paper is well-written. There are only a few typos.

“By integrate a large number of diverse retrieval systems into BEIR, creating such diverse pools becomes significantly simplified.”  -> integrating


**Correctness:**

The experimental setup is generally ok. In results for efficiency, it seems that they report execution time for running the systems only once. It would be better to run them multiple times (e.g., 3) and report average.

**Documentation:**

The authors use only publicly available datasets. They provide URLs to the datasets and an API to use the benchmark.

**Ethics:**

There is no ethical issue.

**Relation To Prior Work:**

The authors provide a discussion about related work. I think they have a solid contribution.

**Summary And Contributions:**

The authors introduce BEIR benchmark for zero-shot evaluation of information retrieval (IR) models. BEIR consists of 18 datasets covering 9 different IR tasks. The tasks include fact checking, citation prediction, duplicate question retrieval, argument retrieval, news retrieval, question answering, tweet retrieval, bio-medical IR, and entity retrieval. The authors also evaluate 10 different retrieval methods. They also provide an API to use the benchmark. They claim that they will provide a leaderboard for the models use BEIR. The leaderboard shared by the authors is a Google Spreadsheet showing ranking of IR systems.

---

> ### Author Response · Authors · 2021-09-29
> **Thank you for the feedback**
>
> Thank you for your review for summarizing our paper’s strengths and mentioning insightful weaknesses and pointing out few typos. The BEIR Python package computes several metrics (Precision, Recall, MRR, MAP, nDCG) and lets you add different cut-off values in (1, 3, 5, 10, 100, 1000). The nDCG@20 evaluation metric and model scores could easily be included within the final draft.
>
> In the paper, we decided to focus on nDCG@10 and Recall@100 (Table 9 in the appendix). We provided the motivation for nDCG in Section 3.3 (Lines 177-185). We selected the cut-off value as 10 for nDCG following previous TREC conferences which used nDCG@10 as their primary evaluation metric, such as TREC Deep Learning 2019 [1] and 2020 [2], TREC COVID up to round 4 [3], and TREC 2019 Decision Track [4]. A different cut-off, e.g. of 5 or 20, would also have been possible, but we did not observe large differences in the results. Therefore, we decided to stay in line with the same evaluation as the TREC Deep Learning 2019/2020 that used nDCG@10.
>
> We agree Google Spreadsheet is a temporary solution for hosting the leaderboard. We would like to design a website in the future that could host the BEIR leaderboard. Thanks for suggesting the idea that people can submit their systems outputs, and it can be automatically used to update the leaderboard. This is a nice idea, which we hopefully can add in the near future to the leaderboard.
>
> Regarding Correctness: Yes, we reported the execution time for running the systems only once. However, we reported the average latency time on a subset of the DBPedia dataset (with 1 Million documents) and repeated the experiment across 400 questions for each model. We will update the final paper and report average latency times. So far, we did not observe large differences between two runs.
>
>
> [1] https://arxiv.org/abs/2003.07820
>
> [2] https://arxiv.org/abs/2102.07662
>
> [3] https://ir.nist.gov/trec-covid/round4.html
>
> [4] https://trec.nist.gov/pubs/trec28/papers/OVERVIEW.D.pdf

---

### Official Review · Reviewer_9eWA · 2021-09-21
**Well constructed IR Benchmark**

**Rating:** 8
**Confidence:** 5
**Correctness:** The claims made in the paper are corr…
**Clarity:** The paper is clearly written and is a…

**Strengths:**

1) This paper deals with a wide variety of text retrieval tasks -- with varying vocabulary, domains, and annotation sparsities. This makes it a first diverse benchmark for text ranking.

2) One of the major aspects of this benchmark is that it is easy to use and experiment with. Even if the benchmark pitches itself as targetted towards zero-shot learning, BEIR could infact be used to benchmark standard ranking models in the fine-tuning regime as well.

3) The number of approaches considered is both sufficient as well as diverse to justify "prepping" of the benchmark. Most of the sparse and dense approaches are well known and provide some insight into the task of zero-shot transfer.






**Weaknesses:**


1) The difference between retrieval and ranking performance is not argued in detail. In IR, both retrieval and ranking are crucial for the understanding of text-ranking models. The distributions are typically different for retrieval and ranking tasks. Specifically, while the retrieval model has to differentiate between a large set of non-relevant items and the actual relevant item/s, the ranking model has to differentiate between partially relevant items and the actual relevant item/s. Typically, most of the models are trained using the ranking data since it has better hard negatives. This also might result in overfitting the task data with very low zero-shot performance. It is not clear in which regime are the models presented in the paper trained on. The authors should clarify this point and maybe reflect on the deficiencies of training on task-specific hard negatives on the results.

2) Although the paper does a great job in curating a number of retrieval tasks, it misses out on the bread-and-butter long document retrieval task for the Web domain. The reasons for leaving this out in also not discussed. Possibly ClueWeb09 and TREC-DL datasets could be good candidates. Web datasets are often accompanied by a higher degree of query underspecification and hence are more complex than other included tasks. Granted that the news datasets also contain similar queries, but Web datasets exhibit other complexities like redundancy and high content variability.

3) The datasets considered do not come with an accompanying corpus. Since the focus is on retrieval models and re-ranking, accompanying corpora would help provide abilities for superior modeling


**Additional Feedback:**

The authors should re-visit the word overlap experiment to show task complexity/overlap. Rather use the real estate to show transfer performance when the models are trained on a task (apart from MSMarco). The word overlap experiment is slightly misleading and doesnt provide extra intuition about task complexity.

**Documentation:**

The datasets are well documented and easy to reproduce.

**Ethics:**

There are no ethical concerns because the authors use public datasets.

**Relation To Prior Work:**

The paper covers most of the important related works and is well contextualized.

**Summary And Contributions:**


This paper presents a benchmark for text retrieval tasks -- a central task in Information Retrieval (IR). This work selects 18 already existing datasets from diverse domains and shows the effectiveness of recently-proposed (and now standard) transformer-based retrieval models. The datasets in BEIR cover a wide variety of domains and task complexities. The strongest point of the paper is the large variety of models used to showcase retrieval and ranking performance in a transfer learning setting. In fact, the task complexity is shown by the non-transferability of the models from the MSMarco task to others. Or more specifically, most of the neural retrieval models perform worse than the simple yet robust non-parameterized BM25 on zero-shot transfer.

---

> ### Author Response · Authors · 2021-09-29
> **We are happy to hear that you like the benchmark**
>
> Thank you that you liked the benchmark. We find it encouraging to hear that you find the benchmark diverse, and easy-to-use to experiment with.
>
> Regarding the mentioned points:
> 1) **Differences between retrieval and ranking performance:** Thank you for mentioning this discussion topic. We will add the missing discussion of how the models were trained to the final version of the paper. DocT5query and DeepCT were trained for document expansion & term re-weighting. CrossEncoder (MiniLM) and SPARTA were trained with ranking data. The dense approaches (ANCE, TAS-B) and ColBERT were trained with a mixture: both ranking data and random in-batch negatives.
> 2) **Long document retrieval tasks:** As mentioned in the appendix (B Limitations of the BEIR benchmark), the current datatsets have rather short documents of a few paragraphs. This is mainly due to the 512 token limit of the tested models: Long-document retrieval must be structured differently if transformer models should be used. Different strategies to split documents into shorter sections and to aggregate them to a final score must be tested and compared (like sum, mean, max of the individual scores). We thought this is out-of-scope for the current version as it would require a quite different model and evaluation setup. But we find long-document-retrieval highly relevant and want to include it in  the future to BEIR, including different strategies to split documents and aggregate scores. **TREC-DL and ClueWeb Datasets:** Thank you for suggesting TREC-DL and ClueWeb09 as retrieval tasks for the web domain. We left out TREC-DL  2019 / 2020 (Passages) as it uses queries and documents from the MS MARCO dataset, for which most models (except DPR) were trained and optimized for. Here, we noticed a strong correlation between MS MARCO Dev and TREC DL 2019 / 2020 performance. Instead, we decided to focus on the out-of-domain evaluation. We left out ClueWeb09 as it consists of over 1 Billion documents. Unfortunately, we didn’t have enough computational resources to evaluate all approaches on this large dataset. For example, evaluating docT5query on this would require 140 days on a TPUv3 costing about 90k USD on a preemptible TPU. As the dataset contains long documents, which need to be splitted into multiple sections, it would further increase the computation complexity.
> 3) **Datasets considered do not come with an accompanying corpus:** Could you elaborate on this point? Do you mean a training corpus? If yes, then most datasets we chose in BEIR don’t necessarily come with a training corpus (if they have a training corpus, it is mentioned in Table 1). But we would find it interesting to explorer which datasets could be used for training to improve zero-shot information retrieval. So far we had a large focus on MS MARCO, but there might be other (better) datasets.  The presented benchmark could help to compare different training corpora. But one should be careful to not overfit on the selected tasks.

---

### Official Review · Reviewer_J1ub · 2021-09-23
**This is a paper that benchmarks several IR techniques on 18 publicly available datasets**

**Rating:** 5
**Confidence:** 5
**Clarity:** Yes

**Strengths:**

Extensive analysis of several IR models using both in-domain and out-of domain datasets.
Discussed the issues of annotation selection bias
A nice framework to evaluate many IR techniques using many datasets.

**Weaknesses:**

All these experimental findings are already knew. The authors fails to suggest any new insights.

**Additional Feedback:**

Nice python package. I hope you will actively maintain the repo.

**Correctness:**

The authors major claim is that the framework is easily adaptable for other IR tasks and datasets but there is no discussion about it in the paper.

**Documentation:**

Yes, nicely documented.

**Relation To Prior Work:**

Not applicable

**Summary And Contributions:**

The authors benchmark 10 IR techniques on 18 publicly available datasets. Their findings conclude that BM25 is a robust baseline which is a known phenomenon from many other studies in IR evaluations. The authors define 3 criteria for selecting 18 datasets which are also very generic and nothing out of ordinary. However, the result and analysis section is a nice read.

---

> ### Author Response · Authors · 2021-09-29
> **The main contribution is a well crafted and re-usable benchmark**
>
> Thank you for your review. We would like to address your concern that all experimental findings are already known and the paper fails to suggest any new insights.
>
> The main contribution of this work is the proposal of a standardized benchmark for the zero-shot evaluation of retrieval systems. It tests retrieval systems on a diverse range of tasks and domains. As highlighted in Section 2 (Lines 72-81), previous (standardized) benchmarks included a narrow evaluation setup, either on their task (e.g. MultiReQA just focusing on question-answering) or on their retrieval corpus (e.g. KILT just retrieves from Wikipedia). BEIR overcomes this shortcoming and provides an easy-to-use evaluation framework for new retrieval methods.
>
> Evaluation in the existing literature is often narrow and in-domain, making it difficult to infer conclusions on generalization to other retrieval tasks or domains. For example, DeepCT [1] was evaluated in-domain on MS MARCO & TREC CAR datasets, showing a significant outperformance against BM25 and doc2query, leading the authors to the conclusion that “DeepCT [...] better estimates term importance for first-stage bag-of-words retrieval systems”. As shown in the paper, the conclusion does not hold true for generalization in a zero-shot setting as DeepCT underperforms BM25 on all 18 other retrieval tasks. Similarly, DPR [2] was evaluated on only question-answering datasets, leading to the conclusion that “DPR can [...] replace the traditional sparse retrieval component”. As shown, DPR underperforms BM25 significantly for all datasets (except for NQ, on which it was trained), also on question-answering datasets from other domains (FiQA, BioASQ). Finally, SPARTA [3] performs better than docT5query [4] in-domain on MS MARCO dev (+1.3 points NDCG@10), but docT5query is better than SPARTA on 17 out of 18 zero-shot evaluated datasets.
>
> Further, evaluation in earlier literature is heterogeneous in terms of used datasets and evaluation metrics: Depending on the paper, authors report either MRR@10, NDCG@10, NDCG@20, or Top-k-Accuracy. This mixture of evaluation metrics makes it challenging to compare systems on an equal footing.
>
> A standardized benchmark for IR systems helps to: 1) fairly compare different retrieval approaches across domains & tasks, 2) identify promising architectures that will perform well across tasks, and 3) steer innovation towards more robust retrieval systems. We find it encouraging that BEIR is already used by new retrieval systems (e.g. SPLADE-v2 [6]) to compare against existing approaches.
>
> The second contribution of this paper is the broad evaluation of systems on a large set of 18 datasets. To our knowledge, it is the first paper that evaluates recent diverse retrieval systems in a fair setup on many diverse tasks from different domains. This generated many new, smaller insights which we have not seen previously: 1) BM25 is still a strong system even when compared to neural systems from early last year (2020/21), 2) In-domain retrieval performance (e.g. on MS MARCO dev) has little correlation to out-of-domain retrieval performance, 3) Strong generalization differences between retrieval systems even when using a comparable setup (e.g. DeepCT vs. docT5query and TAS-B vs. DPR), 4) Hole@k analysis on TREC COVID  for different approaches, 5) Many individual systems insights as mentioned above.
>
> [1] Zhuyun Dai and Jamie Callan. 2019. Context-aware sentence/passage term importance estimation for first stage retrieval. arXiv preprint arXiv:1910.10687.
>
> [2] Vladimir Karpukhin, Barlas Oğuz, Sewon Min, Patrick Lewis, Ledell Wu, Sergey Edunov, Danqi Chen, and Wen-tau Yih. 2020. Dense passage retrieval for open-domain question answering. In Proceedings of the 2020 Conference on Empirical Methods in Natural Language Processing (EMNLP).
>
> [3] Tiancheng Zhao, Xiaopeng Lu, and Kyusong Lee. 2021. Sparta: Efficient open-domain question answering via sparse transformer matching retrieval. In Proceedings of the 2021 Conference of the North American Chapter of the Association for Computational Linguistics: Human Language Technologies.
>
> [4] Rodrigo Noguiera and Jimmy Lin. 2019. From doc2query to docTTTTTquery. Online preprint. https://cs.uwaterloo.ca/~jimmylin/publications/Nogueira_Lin_2019_docTTTTTquery-v2.pdf
>
> [5] Xiong, Lee, Chenyan Xiong, Ye Li, Kwok-Fung Tang, Jialin Liu, Paul Bennett, Junaid Ahmed, and Arnold Overwijk. 2021. Approximate nearest neighbor negative contrastive learning for dense text retrieval. In Proceedings of International Conference on Learning Representations.
>
> [6] Thibault Formal, Carlos Lassance, Benjamin Piwowarski, and Stéphane Clinchant. 2021. SPLADE v2: Sparse Lexical and Expansion Model for Information Retrieval. arXiv preprint arXiv:2109.10086.

---

### Decision · Program_Chairs · 2021-10-10

**Decision:**

Accept

**Comment:**

This paper presents a benchmark for text retrieval tasks, using 18 already existing datasets from diverse domains and task complexities, and covering a large variety of models used to showcase retrieval and ranking performance especially in a transfer learning setting. Overall, reviewers thought the work was strong in the diversity of included tasks, models, and usability; reviewers also praised the relevance with respect to zero-shot approaches. Concerns included overall novelty, distinction between relevance and ranking, and the fact that models were trained on different corpora at times. However, the reviewers ultimately have found this paper to be an extremely strong benchmark that will be immediately useful in the field.